# Effects of Whole-Body Adenylyl Cyclase 5 (*Adcy5*) Deficiency on Systemic Insulin Sensitivity and Adipose Tissue

**DOI:** 10.3390/ijms22094353

**Published:** 2021-04-21

**Authors:** Sebastian Dommel, Anne Hoffmann, Claudia Berger, Matthias Kern, Nora Klöting, Aimo Kannt, Matthias Blüher

**Affiliations:** 1Medical Center, Medical Department III—Endocrinology, Nephrology, Rheumatology, University of Leipzig, 04103 Leipzig, Germany; sebastian.dommel@medizin.uni-leipzig.de (S.D.); claudia.berger@medizin.uni-leipzig.de (C.B.); nora.kloeting@medizin.uni-leipzig.de (N.K.); 2Helmholtz Institute for Metabolic, Obesity and Vascular Research (HI-MAG) of the Helmholtz Zentrum München at the University of Leipzig and University Hospital Leipzig, 04103 Leipzig, Germany; anne.hoffmann@helmholtz-muenchen.de (A.H.); matthias.kern@helmholtz-muenchen.de (M.K.); 3Fraunhofer Institute for Translational Medicine and Pharmacology ITMP, 60596 Frankfurt am Main, Germany; aimo.kannt@itmp.fraunhofer.de; 4Experimental Pharmacology, Medical Faculty Mannheim, University of Heidelberg, 68167 Mannheim, Germany; 5Sanofi Diabetes Research and Development, 60596 Frankfurt am Main, Germany

**Keywords:** adipocyte size, insulin resistance, metabolism, gene expression, *Adcy5* knockout, running activity

## Abstract

Genome-wide association studies have identified adenylyl cyclase type 5 (*ADCY5*) as candidate gene for diabetes-related quantitative traits and an increased risk of type 2 diabetes. Mice with a whole-body deletion of *Adcy5 (Adcy5*^–/*–*^*)* do not develop obesity, glucose intolerance and insulin resistance, have improved cardiac function and increased longevity. Here, we investigated *Adcy5* knockout mice (*Adcy5*^–/*–*^) to test the hypothesis that changes in adipose tissue (AT) may contribute to the reported healthier phenotype. In contrast to previous reports, we found that deletion of *Adcy5* did not confer any physiological or biochemical benefits. However, this unexpected finding allowed us to investigate the effects of *Adcy5* depletion on AT independently of lower body weight and a metabolically healthier phenotype. *Adcy5*^–/*–*^ mice exhibited an increased number of smaller adipocytes, lower mean adipocyte size and a distinct AT gene expression pattern with midline 1 (*Mid1*) as the most significantly downregulated gene compared to control mice. Our *Adcy5*^–/*–*^ model challenges previously described beneficial effects of *Adcy5* deficiency and suggests that targeting Adcy5 does not improve insulin sensitivity and may therefore limit the relevance of ADCY5 as potential drug target.

## 1. Introduction

Adenylyl cyclase type 5 (*ADCY5*) has been identified as candidate gene for an increased risk to develop type 2 diabetes (T2D) in a genome-wide association study [1]. Carriers of the high risk single nucleotide polymorphism (SNP) rs11708067 within the *ADCY5* gene have lower *ADCY5* mRNA expression in islets associated with fasting hyperglycemia and higher T2D risk [1,2]. In addition, *ADCY5* genetic variants have been associated with lower birth weight [3], gestational diabetes [4], and parameters of insulin sensitivity in children with obesity [5]. However, associations between the *ADCY5* genotype are heterogeneous and range from being protective against T2D [1,6,7] to increasing the diabetes risk [1,2,5,8,9]. In three cases of exceptional inherited longevity, exome sequencing revealed *ADCY5* as candidate gene [10]. Moreover, mice with a targeted disruption of *Adcy5* have a ~30% extended life span [11]. This observation may be explained by the phenotype of *Adcy5* knockout mice (*Adcy5*^–/–^), which share a phenotype similar to that caused by calorie-restricted wild-type mice [11] or mice with an adipose-tissue-targeted disruption of the insulin receptor [12].

*Adcy5*^–/–^ mice are protected against high-fat diet-induced obesity, insulin resistance [13], heart failure [14,15], myocardial apoptosis [16] and have reduced whole-body oxidative stress [11]. Therefore, ADCY5 has been proposed as a target for the treatment of obesity, diabetes and cardiovascular diseases [13].

Based on these data, we hypothesized that adipose tissue (AT) may represent an important tissue linking the effects of reduced *Adcy5* expression to the observed metabolically healthy phenotype in *Adcy5*^–/–^ mice. In a first cross-sectional human study, we found that adipose tissue *ADCY5* expression positively correlates with BMI, fat mass and parameters of fat distribution in humans [17]. We therefore aimed to investigate the effects of reduced whole-body *Adcy5* expression on body composition, fat distribution, adipocyte morphology and global adipose tissue gene expression.

## 2. Results

### 2.1. Adcy5^–/–^ Mice Are Not Protected against Obesity

Under CD conditions *Adcy5*^–/–^ and Ctrl mice of both sexes exhibited indistinguishable body weight gain until the age of 30 weeks (Figure 1A,B). In male mice, a 20-week HFD challenge started at 6 weeks of age caused obesity without significant differences between the body weight curves of *Adcy5*^–/–^ and Ctrl mice (Figure 1B). We found that female Ctrl mice did not respond as male mice to HFD and gained significantly less weight than female *Adcy5*^–/–^ mice (Figure 1A). Starting from 10 weeks until the end of HFD, female *Adcy5*^–/–^ mice exhibited up to 30% higher body weight gain compared to Ctrl mice (Figure 1A). Since we did not use littermate controls, we performed a Mouse Genome Scan SNP analysis to exclude genetic background effects as causative for the observed weight differences. Genome scans revealed a 99.21% identity between our Ctrl and *Adcy5*^–/–^ mice.

Except for a slight increase in brown adipose tissue (BAT) in *Adcy5*^–/–^, body composition in females fed on CD was comparable for both genotypes (Figure 1C). On the other hand, *Adcy5*^–/–^ mice gained significantly more epigonadal white adipose tissue (eWAT) and inguinal WAT (ingWAT) upon HFD (Figure 1C, Appendix A). Accordingly, relative lean mass was significantly lower in female *Adcy5*^–/–^ compared to controls under both diets (Appendix A). Observed body weight differences in females fed on HFD could not be explained by differences in body length (Appendix A) or by a significantly increased food intake, despite *Adcy5*^–/–^ females tending to feed more than their Ctrl (Appendix A). In contrast to females, male *Adcy5*^–/–^ and Ctrl mice did not differ in their response to HFD (Figure 1B), but *Adcy5*^–/–^ mice had lower eWAT and ingWAT mass under CD (Figure 1D). Male *Adcy5*^–/–^ mice showed a trend for lower body fat mass and higher relative lean body mass compared to Ctrl animals (Appendix A). Body length was negligibly different for both genotypes upon HFD (Appendix A). Surprisingly, food intake was not different between female *Adcy5*^–/–^ and Ctrl mice for both diets, (Appendix A) whereas male *Adcy5*^–/–^ mice consumed significantly less under HFD conditions (Appendix A).

Taken together, *Adcy5*^–/–^ mice were not protected against HFD-induced obesity. The higher degree of adiposity in female *Adcy5*^–/–^ mice is most likely explained by insufficient response to HFD in Ctrl mice.

### 2.2. Adcy5 Deletion Effects on Glucose Tolerance, Insulin Sensitivity and Serum Parameters

To detect potential disorders in glucose metabolism resulting from the *Adcy5* knockout, we performed intraperitoneal glucose tolerance tests (ipGTTs). Under CD conditions, female *Adcy5*^–/–^ mice responded with a higher increase in blood glucose 15 min after glucose injection (2 g glucose per kg body weight). Regarding the area under the curve (AUC) for both diets, *Adcy5*^–/–^ females exhibited a trend to develop glucose intolerance (Figure 2A). On the contrary, CD male *Adcy5*^–/–^ mice exhibited lower circulating glucose concentrations after the intraperitoneal glucose challenge, which did not result in significantly lower AUC glucose during the ipGTT course (Figure 2B). Challenged with a HFD this trend became inverted, resulting in higher AUC glucose in *Adcy5*^–/–^ male mice (Figure 2B).

Fasting blood glucose levels were not different between female *Adcy5*^–/–^ and control mice fed on CD, but there was a ~20% increase for *Adcy5*^–/–^ upon HFD, which was not significant (Table 1). Female *Adcy5*^–/–^ mice were further characterized by significantly lower HbA1c levels under CD (Table 1). Interestingly, lean male *Adcy5*^–/–^ mice exhibited significantly lower HbA1c levels, fasting blood glucose and C-peptide concentrations compared to Ctrl (Table 2). However, these differences were not observed under HFD conditions (Table 2).

In hyperinsulinemic-euglycemic clamp studies under CD conditions, we did not find significantly impaired whole-body insulin sensitivity in *Adcy5*^–/–^ compared to Ctrl mice (Figure 2C–F).

Stimulated with 11 and 22 mmol glucose, pancreatic islets of *Adcy5*^–/–^ females secreted lower amounts of insulin into medium compared to islets of Ctrl animals (Figure 2G). There were no genotype differences in glucose-stimulated insulin secretion at other glucose concentrations in females (Figure 2G) or at any glucose concentration in males (Figure 2H). In both sexes, total insulin content per islet after glucose stimulation was not significantly different between Ctrl and *Adcy5*^–/–^ except for male mice stimulated with 5.5 mmol glucose (Figure 2I,J).

Regarding serum lipids, *Adcy5^–/–^* females were characterized by significantly lower total and HDL-cholesterol compared to Ctrl under CD (Table 1). In contrast, total and HDL-cholesterol were significantly higher in female *Adcy5*^–/–^ after HFD. In line with females, male *Adcy5^–/–^* had higher total and HDL-cholesterol serum concentrations under HFD, and significantly lower LDL-cholesterol serum levels under CD (Table 2). Leptin and adiponectin serum concentrations were not different between male *Adcy5^–/–^* and Ctrl mice under both dietary interventions (Table 2). In female *Adcy5^–/–^* mice we found higher leptin levels after HFD (Table 1). Additionally, microarrays performed with RNA isolated from eWAT of mice on HFD did not show differentially expressed pro-inflammatory genes such as *Il6, Tnfa, Nfkb1* or *Ccl2* between *Adcy5^–/–^* and control animals of both sexes (Figure 4).

In summary, lean *Adcy5^–/–^* mice exhibited lower HbA1c levels compared to controls independent of their sex. In contrast, glucose tolerance was in both sexes not significantly different between the genotypes (Appendix A), but *Adcy5*^–/–^ mice showed a trend for developing glucose intolerance under HFD. Furthermore, insulin sensitivity was not affected by deletion of *Adcy5*. Compared to Ctrl, 20 weeks of HFD challenge resulted in higher total and HDL-cholesterol serum levels for *Adcy5*^–/–^ mice of both sexes.

### 2.3. Consequences of Adcy5^–/–^ on AT Morphology

Histological analyses of ingWAT and eWAT revealed that mean and maximal adipocyte sizes were significantly lower in *Adcy5^–/–^* compared to control mice of both sexes under CD (Figure 3). Particularly, the subfraction of small adipocytes was enriched in *Adcy5^–/–^* compared to Ctrl mice both in ingWAT (Figure 3A–D) and eWAT (Figure 3E–H). Smaller adipocytes in *Adcy5^–/–^* vs. Ctrl mice were further reflected by lower mean adipocyte areas per given analysis area of whole adipose tissue sections (Figure 3I,J). We did not find evidence for differences in adipose tissue immune cell infiltration between *Adcy5^–/–^* and Ctrl animals. The AT expression of immune cell markers CD68, CD40, CD11c and CD163 was not different between *Adcy5^–/–^* and Ctrl mice indicated by non-significant gene expression changes in microarray investigations (Figure 4A–D). HFD challenge resulted in the formation of larger adipocytes in female *Adcy5^–/–^* compared to Ctrl mice (Appendix A). In contrast, male *Adcy5*^–/–^ mice revealed smaller adipocytes under HFD challenge (Appendix A).

In summary, *Adcy5* knockout affected AT morphology and was characterized by smaller adipocytes except for females fed on HFD, but not by differences in AT inflammation markers.

### 2.4. Adcy5^–/–^ Mice Are Characterized by a Distinct EWAT Gene Expression Signature

Global microarray gene expression analyses in eWAT of CD mice revealed 64 (females) and 17 (males) differentially expressed genes (*p*-value < 0.01; |FC| ≥ 2) between *Adcy5^–/–^* and Ctrl animals (Figure 4A–D; Appendix A). In both sexes, we found midline 1 (*Mid1*) as most significantly (FDR < 0.05) downregulated gene (logFC = –2.14 in females, males: –2.30) in *Adcy5^–/–^* vs. Ctrl mice. In female *Adcy5^–/–^* mice sodium/potassium-transporting ATPase subunit alpha-3 (*Atp1a3*) was one of the most significantly upregulated genes (logFC = 2.02) (Figure 4A). Interestingly, several genes of lipid (e.g., *Elovl6* [logFC = 2.37], *Fasn* [logFC = 1.40]) and glucose metabolism (e.g., *Gpd1* [logFC = 1.65], *Slc2a5* [logFC = 1.61]) were significantly higher expressed in AT of *Adcy5^–/–^* compared to Ctrl mice (Appendix A). According to KEGG analysis of our microarray data obtained for AT gene expression analyses, the most upregulated genes contributed to metabolic pathways like oxidative phosphorylation, citrate cycle, thermogenesis or glycolysis and gluconeogenesis in females. Interestingly, we found pathways relevant to the pathogenesis of Parkinson’s, Alzheimer´s, Huntington´s or non-alcoholic fatty liver disease to be upregulated in female *Adcy5^–/–^* mice (Figure 4C).

Global AT gene expression analysis confirmed the downregulation of *Adcy5* expression and demonstrated that no other adenylyl cyclase gene expression was affected by targeting *Adcy5* at the whole-body level (Figure 4E). To sum up, deletion of *Adcy5* was not compensated by higher expression of other adenylyl cyclases. However, *Adcy5^–/–^* mice presented an altered expression of genes involved in metabolic pathways.

### 2.5. Adcy5^–/–^ Affects Activity and Energy Expenditure

In previously characterized *Adcy5* knockout mice, higher energy expenditure (EE) was reported together with beneficial effects on body fat mass, glucose metabolism and insulin sensitivity [13]. We therefore measured daily food intake, energy expenditure (EE) and running wheel activity over 48 h (Figure 5 and Appendix A). Except for male mice under HFD, food intake was not significantly different between *Adcy5^–/–^* and Ctrl mice in both sexes and diet interventions (Appendix A). Female *Adcy5^–/–^* mice of both diets consumed significantly less oxygen and had a significantly lower EE compared to control mice under HFD (Figure 5A,B,D,F,G,I). Male *Adcy5^–/–^* and control mice were not significantly different in these parameters of oxygen consumption and energy expenditure (Figure 5K,L,N,P,Q,S). RER values were not found to be different in any sex or diet (Figure 5C,H,M,R). Importantly, in both male and female *Adcy5^–/–^* mice, exercise running wheel activity was significantly lower under both CD and HFD conditions (Figure 5E,J,O,T). Reduced exercise wheel activity was the most striking behavioral difference between the genotypes.

## 3. Discussion

Our study was inspired by previous reports that mice with a disruption of *Adcy5* are protected against the development of high fat feeding-induced obesity, glucose intolerance, insulin resistance [13], heart failure [15], have reduced oxidative stress [18] and increased longevity [11]. In addition, deletion of *Adcy5* caused reduced fat mass, particularly in visceral depots and changes in adipose tissue morphology mimicking the effects of caloric restriction [13,18]. Mechanisms linking *Adcy5* deficiency to these health benefits (e.g., decreased glycogen storage) seemed to be so similar to the effects of caloric restriction that *Adcy5* knockout mice have a shortened life span upon calorie restriction [18]. We previously showed that mice with a conditional disruption of the insulin receptor in adipose tissue have increased longevity [12,19] and share similarities to long-lived mice and rats in response to caloric restriction [20,21,22]. Altered AT function is one important aspect linking impaired insulin [12] or insulin-like growth factor-1 (IGF-1) signaling to longevity [23]. Based on these data, we hypothesized that reduced *Adcy5* expression causes changes in AT that contribute to the reported healthier cardio-metabolic phenotype of *Adcy5* knockout mice, which ultimately led to prolonged lifespan. We therefore generated *Adcy5^–/–^* mice with the same targeting strategy as the previously reported *Adcy5* knockout mice [11,13,24].

Unexpectedly, we found that *Adcy5^–/–^* mice of both sexes were not protected against obesity and impaired glucose metabolism. Moreover, the previously described effects of *Adcy5* deletion on improved insulin sensitivity, increased oxygen consumption, respiratory exchange ratio, and energy expenditure could not be confirmed in our model. In our model, ablation of *Adcy5* at the whole-body level did not protect mice against obesity under high-fat diet conditions. Our model was further complicated by the observation that only female wild-type controls did not respond as expected to HFD with the development of obesity. The inappropriate response to HFD in control mice led to significantly higher body weight, impaired glucose tolerance and insulin sensitivity only in female *Adcy5^–/–^* mice.

It is well known that males and females are differentially susceptible to diet-induced obesity (DIO). Female mice have been shown to develop obesity later and to a lesser extent than males [25]. Comparing male mice with ovariectomized and non-ovariectomized females, Stubbins et al. found that ovary removal eliminated protection of female mice against DIO [26]. Here, we showed that females lacking *Adcy5* developed severe obesity under HFD, comparable to male mice. In mice, *Adcy5* is most highly expressed in subcutaneous fat and the ovary [27]. The lack of *Adcy5* results in decreased levels of cAMP. Beside other effects, cAMP was shown to highly increase aromatase expression, the key enzyme in estrogen production, in stromal cells [28]. In summary, our *Adcy5* knockout mice are similar to estrogen-lacking ovariectomized mice described by Stubbins et al. [26]. Unfortunately, we were not able to assess circulating estrogen concentrations in our model and could therefore only speculate that genotype differences in estrogen levels may contribute to the distinct phenotype of our female mice under HFD.

For our studies, littermate controls from the *Adcy5^–/–^* breeding were not available so we used wild-type mice of the same C57BL/6NTac background as controls. To investigate whether differences in the genetic background between *Adcy5^–/–^* and control mice may underlie the observed differences in HFD response of female mice, we performed Genome Scan SNP analyses. These analyses showed a 99.21% identity between Ctrl and *Adcy5^–/–^* animals, suggesting that the control mice sufficiently represent the background of *Adcy5^–/–^* mice. The analysis revealed that both *Adcy5*^–/–^ and Ctrl mice are homozygous wild-type for nicotinamide nucleotide transhydrogenase (*Nnt*), a gene that we and others found to be associated with different DIO-susceptibility in C57BL/6 substrains [29,30]. Additionally, we previously showed that only four heterogeneous sex-specific genetic variants underlie the differences in HFD-induced weight gain between the otherwise genetically identical C57BL/6NTac and C57BL/6JRj substrains [31]. Since our Adcy5 knockout on C57BL/6NTac background differed from previously described ones, e.g., 129/Sv [18], C57BL/6J [13], or 129/SvJ-C57BL/6 mixed [14], we could not exclude that even small genetic background differences may have caused our contrasting findings compared to previous *Adcy5* deletion models (reviewed in [16]). Importantly, the comparison of our model with previous data [13] suggested that interactions between *Adcy5* deletion and the genetic background may play a previously unrecognized role determining the phenotype. If this assumption is true, it would turn ADCY5 into a less likely therapeutic target for the treatment of obesity and diabetes.

Individual variation in *Adcy5* interaction with the genetic background may also explain the heterogeneous findings from genome-wide association studies and candidate gene approaches reporting different directions of the associations of ADCY5 genetic variants with obesity, diabetes-related quantitative traits and the risk of type 2 diabetes [1,2,5,8,9].

Additionally, the different dietary compositions between our HFD and HFDs used in previous studies could explain the various phenotypes [32,33].

However, the unexpected lack of a difference in body weight, fat mass, energy expenditure, glucose metabolism and insulin sensitivity between our male *Adcy5^–/–^* and control mice could also be an advantage, because it allowed us to investigate the effects of *Adcy5* depletion on AT independently of lower body weight and a metabolically healthier phenotype. We therefore studied AT morphology and gene expression signatures to test the hypothesis that ablation of *Adcy5* may have weight-independent effects on AT. In humans, we showed that AT *ADCY5* expression is related to obesity and fat distribution, but not with impaired glucose metabolism and T2D [17]. In addition, altered *ADCY5* expression in AT does not seem to underlie the association between the ADCY5 SNP rs11708067 and increased T2D risk [17].

It has been previously suggested that ACDY5 couples glucose to insulin secretion in human islets because reduced *ACDY5* expression was associated with impaired insulin secretion after a glucose challenge [2]. Accordingly, islets of our *Adcy5*-deficient females exhibited lower glucose-stimulated insulin secretion into the medium after stimulation with high glucose concentrations. Importantly, we could not confirm these potential *Adcy5* knockout-related effects at lower glucose concentrations in females or at any given glucose level in male mice. In addition, euglycemic-hyperinsulinemic clamp studies did not reveal an effect of *Adcy5* deficiency on systemic insulin sensitivity. In contrast, we found just minor and non-significant effects on glucose tolerance upon depletion of *Adcy5* in our model. In male *Adcy5^–/–^* mice, HOMA-ß was significantly higher under CD conditions, further supporting an unexpectedly better beta cell function despite reduced *Adcy5* expression.

Except for females on HFD, *Adcy5^–/–^* mice exhibited an increased number of smaller adipocytes and lower mean adipocyte size. Smaller adipocyte size is a consistent finding compared to previous studies [13], suggesting that *Adcy5* ablation affects AT independently of reduced fat mass. An increased number of smaller adipocytes may link *Adcy5* knockout to the previously reported effects on longevity and reduced diabetes risk, because small adipocytes are strongly related to BMI-independent retained insulin sensitivity [34]. Both animal [12] and human studies [35] have suggested that protection against adipocyte hyperplasia is associated with a lower risk of hyperglycemia and type 2 diabetes. Nevertheless, in our model the smaller adipocyte size did not translate into better insulin sensitivity. In the context of glucose metabolism, we found HbA1c levels marginally reduced in lean *Adcy5*^–/–^ mice of both sexes (~4% in males, ~6% in females), whereas fasting blood glucose levels tended to be enhanced after HFD challenge (~17% in males, ~23% in females). In contrast to males, *Adcy5*^–/–^ females developed larger adipocytes under HFD and higher circulating leptin levels. We found a potentially important diet–genotype interaction in distinct parameters of cholesterol metabolism. Since female *Adcy5*^–/–^ mice had significantly lower total and HDL-cholesterol serum concentrations under chow fed conditions, but significantly higher levels of these lipid metabolism parameters after HFD challenge, our data suggest a previously unrecognized role of Adcy5 in the regulation of cholesterol metabolism as demonstrated for Adcy1. In THP-1 foam cells, Adcy1 works as a signaling switch for the apolipoprotein A1-mediated cholesterol efflux pathway. The authors concluded that cAMP, produced by Adcy1, may regulate effectors for cholesterol exocytosis to remove excessive cellular lipids [36]. Additionally, we observed surprisingly high FFA serum concentrations in both genotypes. However, our data reflected previously reported FFA levels in mice after 16 h of fasting [37].

Using global gene expression analyses, we identified *Mid1*, *Atp1a3* and several genes of glucose and lipid metabolism (e.g., *Elovl6*, *Gpd1*, *Gpd2*, *Slc2a5*) as the most significantly differentially expressed genes between female *Adcy5^–/–^* and Ctrl mice. *Mid1* is an E3 ubiquitin ligase of the Tripartite Motif (TRIM) subfamily of RING-containing proteins and is therefore also known as *Trim18* [38]. *Mid1* is involved in several essential biological processes, especially during embryonic development. Binding to Mid1-interacting protein 1 (Mid1ip1, also known as Mig12), resulting Mid1/Mig12 heterodimers stabilizes microtubules in cell division and migration [39]. Interestingly, Mig12 depletion was shown to activate the master regulator of fatty acid oxidation or synthesis, AMP-activated protein kinase (Ampk) [40,41,42]. Taken together, reduced *Mid1* expression in AT of *Adcy5*^–/–^ mice could explain the observed morphological changes in AT, especially under HFD. However, to investigate a potential regulatory function of Mid1-Mig12/Thrsp-Acc1/Acc2 or Mid1-Mig12-Ampk on AT metabolism another study would be necessary.

We further found that both male and female *Adcy5^–/–^* mice were characterized by significantly lower voluntary running wheel activity in metabolic chambers. As mutations in the *Atp1a3* gene are known to be related to dyskinesia, epilepsies and Parkinson’s disease [43] this could be a relevant finding in the context of other reports describing a connection between the loss of *Adcy5* and movement disorders like dyskinesia or autism-like behavior [44,45,46]. Furthermore, we cannot exclude that lower activity in our *Adcy5^–/–^* mice was caused by previously reported movement disorders and contributed to the phenotype differences compared to previous *Adcy5^–/–^* models [11,13,24]. In addition to lower exercise activity, we did not observe any behavioral abnormality in *Adcy5^–/–^* mice. Noteworthy, independent groups reported that mice with a genetic disruption of *Adcy5* exhibited increased repetitive behaviors and sociability deficits similar to autism-like behavior [47]. In humans, mutations of the *ADCY5* gene have been associated with early-onset autosomal dominant chorea and dystonia [45].

In summary, ablation of the *Adcy5* gene did not confer any physiological or biochemical benefits, under neither normal nor high-fat diet challenged conditions. *Adcy5*^–/–^ mice were characterized by considerably lower physical activity. In contrast to previous independent models of *Adcy5* deficiency, we found that targeting ADCY5 did not improve insulin sensitivity.

## 4. Materials and Methods

### 4.1. Animal Studies

All animal studies were approved by the local authorities of the state of Saxony, Germany (ethic approval code TVV60/16, 24.05.2017, Animal Welfare Committee of the Landesdirektion Sachsen, Germany). Mice were housed in pathogen-free facilities in groups of three to five at 22 ± 2 °C on a 12 h light/dark cycle. Animals were bred and kept in the animal laboratories at Leipzig University and were fed a standard chow diet containing 10% calories from fat, 36% from protein and 54% from carbohydrates (CD, Ssniff Spezialdiäten, Soest, Germany, V1324-300). Diet-induced obesity was achieved by feeding a high-fat diet (HFD) containing 54% calories from fat, 20% from protein and 26% from carbohydrates (HFD, Ssniff Spezialdiäten, E15772-34) starting at 6 weeks of age. Both diets were matched with respect to their crude protein content (21.2% vs. 21.6%). All animals had access to food and water ad libitum, except for experiments where a fasting state was required.

### 4.2. Generation of Adenylyl Cyclase Type 5 Knockout Mice

Adenylyl cyclase type 5 knockout (*Adcy5^–/–^*) mice were produced via VelociGene^®^ (Regeneron Pharmaceuticals, INC, Tarrytown, NY, USA) similar to the approach reported by Okumura et al. [24]. *Adcy5* was disrupted by an insertion of a loxP-hUBp-em7-Neo-polyA-loxP cassette into exon 1 (Appendix A). All mice were genotyped by DNA isolation from the tail tip or ear stamp isolated using DirectPCR (Tail) Lysis Reagent (Viagen Biotech Inc., Los Angeles, CA, USA). Primer pairs (biomers.net, Ulm, Germany) are listed in Appendix A. On 2% agarose gels, wild-type (WT) mice showed a 202 bp band, whereas *Adcy5^–/–^* mice were identified by a 152 bp band.

Furthermore, KO efficiency was confirmed in several tissues on both DNA (Appendix A) and mRNA level (Appendix A). Animals were maintained on a C57BL/6NTac background and both sexes were included in all experiments. Due to mice interbreeding, no littermate controls could be generated. Mouse Genome Scan SNP analysis was performed by Taconic (Ms Genome Scan Panel, Taconic Biosciences, Rensselaer, NY, USA) and revealed a 99.21% identity between Ctrl and *Adcy5^–/–^* animals.

### 4.3. Phenotypic Characterization

In this study, 12 mice of each sex and genotype were studied up to age 30 weeks under chow diet (CD) conditions. In addition, a subgroup of 11 female *Adcy5^–/–^* and 14 control mice as well as 13 male *Adcy5^–/–^* and 14 control mice underwent an HFD for 20 weeks starting at age 6 weeks. We recorded body weight weekly and determined whole-body fat and lean mass with the EchoMRI700™ instrument (Echo Medical Systems, Houston, TX, USA) at the study end. Intraperitoneal glucose tolerance tests (ipGTTs; 2 g glucose per kg body weight) were performed at age 25 weeks as previously described [48]. Pancreatic islet isolation and glucose-stimulated insulin secretion experiments were performed as previously described [49]. In brief, islets of all animals were pooled per genotype and sex (*n* = 3). In groups of five, islets were incubated overnight in glucose-free Gibco RPMI 1640 medium (ThermoFisher Scientific, Waltham, MA, USA) supplemented with 5.5 mmol glucose, 10% fetal calf serum and 100 U/mL penicillin and 100 g/mL streptomycin (Sigma-Aldrich, Munich, Germany) at 37 °C and 5% CO_2_. For insulin secretion, RPMI 1640 medium was supplemented with 1.5, 5.5, 11 or 22 mmol glucose, respectively, and islets were incubated for 2 h at 37 °C and 5% CO_2_. Supernatant was collected and islets were sonicated with 0.04 M phosphate buffer. Insulin contents were measured with Ultra Sensitive Mouse Insulin ELISA kit (Crystal Chem, Downers Grove, IL, USA). Hyperinsulinemic-euglycemic clamps were performed at 23 to 25 weeks of age as previously described [50]. In brief, to determine initial GIR rate, mice received insulin continuously infused through an implanted catheter. When blood glucose levels dropped, 20% glucose solution was infused to stabilize blood glucose levels. Short interval blood glucose measurements were performed, and glucose infusion rate adjusted accordingly to reach physiological blood glucose levels of 5.6 mmol/L. When glucose levels were constant for 30 min, the clamp started. In subgroups, whole-body energy metabolism as well as food intake were investigated using an indirect metabolic chamber system at age 25 weeks (HFD) or 29 weeks (CD) (CaloSys V2.1, TSE Systems, Bad Homburg, Germany). In brief, 7 to 10 *Adcy5^–/–^* and Ctrl mice of each sex and diet were housed for 72 h (adaptation time included) in metabolic chambers as previously described (19). Naso-anal length and rectal body temperature were measured at the end of observation period. In EDTA-containing tubes, 20 µL of whole blood was collected for HbA1c analyses (COBAS 7000, Roche, Basel, Switzerland). At age 30 weeks (CD) or 26 weeks (HFD) mice were sacrificed by an overdose of anesthetic (Isofluran, Baxter, Unterschleißheim, Germany). Immediately, inguinal (ingWAT) and epigonadal white adipose tissue (eWAT) as well as liver and brown adipose tissue (BAT) were taken, weighed and stored in liquid nitrogen. For those tissues, relative organ weights were calculated in relation to whole-body weight. ELISA was used to analyze insulin (Mouse Ultrasensitive Insulin ELISA, ALPCO, Salem, NH, USA), C-peptide (Mouse C-peptide ELISA, ALPCO, Salem, NH, USA), leptin (Mouse Leptin ELISA Kit, Crystal Chem Inc., Downers Grove, IL, USA), and adiponectin (Adiponectin (mouse) ELISA Kit, AdipoGen^®^ LIFE SCIENCES, Liestal, Switzerland) serum concentrations using mouse serum according to the manufacturer’s guidelines. In randomly selected subgroups of 3 to 4 mice per sex and genotype, serum protein levels were determined by OLINK proteomics (Uppsala, Sweden).

### 4.4. Adipose Tissue Histology

For histologic investigations, adipose tissue was fixed in 4% buffered formaldehyde for at least 48 h. Afterwards, tissue containing cassettes were rinsed with water and dehydrated in a graded series of 70–100% ethanol followed by ROTI^®^Histol (Carl Roth GmbH, Karlsruhe, Germany) and paraffin. Both, ingWAT and eWAT pads were cut in 5 µm sections and H&E stained afterwards. Systematic analyses with respect to adipocyte size were performed using Keyence BZ-X800 microscope and BZ-X800 Analyzer software (Keyence Corp., Osaka, Japan). At least 1900 (CD) or 2400 (HFD) adipocytes were analyzed from 4 animals for each genotype and sex to determine cell size distribution.

### 4.5. RNA Isolation and Tissue-Specific mRNA Expression

Frozen AT was lysed using QIAzol Lysis Reagent (Qiagen GmbH, Hilden, Germany) and Precellys Homogenizer (Bertin Technologies, Montigny-le-Bretonneux, France). RNA was isolated from tissue homogenates with RNeasy Lipid Tissue Mini Kit (Qiagen). For all other tissues, AllPrep DNA/RNA/Protein Mini Kit (Qiagen) was used for simultaneous purification of DNA, RNA and protein. RNA was reverse-transcribed using Random Primers (Invitrogen, Carlsbad, CA, USA) and Superscript II Reverse Transcriptase (Invitrogen). Quantitative real-time PCR (qRT-PCR) was performed with the LightCycler 480 system and LightCycler 480 SYBR Green I Master (Hoffmann-La Roche AG, Basel, Switzerland) with identical cDNA amounts for each reaction. *Adcy5* mRNA expression was calculated relative to ribosomal protein, large, P0 (*Rplp0* or *36b4*) mRNA. Primers are listed in Appendix A.

### 4.6. Adipose Tissue Microarray Analyses

RNA was isolated from frozen eWAT of *Adcy5^–/–^* (n = 10, 5 females, 5 males) and control (n = 10, 5 females, 5 males) CD mice. Before microarray analysis, RNA integrity and concentration were examined on an Agilent Fragment Analyzer (Agilent Technologies, Palo Alto, CA, USA) using the HS RNA Kit (Agilent Technologies) according to the manufacturer’s instructions. cRNA was prepared from 100 ng of total RNA hybridized to GeneChip Clariom S arrays (ThermoFisher Scientific). Arrays were scanned with a third generation Affymetrix GeneChip Scanner 3000 (ThermoFisher Scientific). Affymetrix GeneChip data were extracted from fluorescence intensities and were scaled to normalize data for inter-array comparison using Transcriptome Analysis Console (TAC) 4.0.2 software.

Raw data were preprocessed applying the oligo Bioconductor R package (v1.50.0, [51]), which performs a deconvolution method for background correction, quantile normalization and uses the Robust Multichip Average (RMA) algorithm for summarization [52]. Quality control of the raw and normalized data was performed using the Biobase (v2.46, [53]) and oligo R Bioconductor packages [54]. Based on the outlier detection test of the arrayQualityMetrics (v3.42, [54]) Bioconductor R package, two male Ctrl and one female *Adcy5*^–/–^ samples were excluded. The differentially expressed genes (DEGs) for the comparisons (i) female *Adcy5^–/–^* vs. female Ctrl and (ii) male *Adcy5^–/–^* vs. male Ctrl were screened using the Linear Models for Microarray data (LIMMA) method (v3.42, [55]). To increase the signal-to-noise ratios array weights were considered. Since only a few genes (36 female; 1 male, see Appendix A) survived multiple testing (FDR < 0.05) applying the Benjamini–Hochberg procedure [56], we set the threshold for identification of DEGs as *p*-value < 0.01 and |FC| ≥ 2. This allowed us to be exploratory about the results, especially in the pathway analysis. A broad gene list functional enrichment analysis for KEGG mouse pathways was performed using Enrichr (www.amppharm.mssm.edu/Enrichr accessed on 19th April 2021) [57,58]. All differentially expressed genes (*p*-value < 0.01) irrespective of FC were used. Microarray data were deposited in the ArrayExpress database at EMBL-EBI (www.ebi.ac.uk/arrayexpress accessed on 19th April 2021) under accession number E-MTAB-9417.

### 4.7. Statistical Analyses

Prism 6.0 software (GraphPad Software Inc., San Diego, CA, USA) was used for graph design. Data are shown as means ± SD or SEM. Multiple t-tests were performed only when no interactions of a grouping variable on a continuous outcome measure were of interest. Otherwise, the residual analysis was conducted to test for the assumption of the two-way ANOVA using the rstatix [59] R package. Here, post hoc tests using pairwise *t*-test were computed for significant interactions only and corrected applying the Bonferroni method [60].

For modeling continuous measurements over a time course of multilevel data we performed repeated-measure two-way ANOVAs (rstatix, [59]) when the data contained a balanced number of replicates or no missing data for some data points. Otherwise, we applied repeated-measure linear mixed-effects models (LMMs) because they account for variability and imbalance in the data. The LMMs were fitted to the data by the formula weight~time*diet*strain+(time|patient ID) applying the lme4 R package [61]. Outliers were assessed by box plot method and normality was assessed using Shapiro–Wilk’s normality test [62]. Homogeneity of variances was assessed by Levene’s test for the two-way ANOVAs and by Mauchly’s test of sphericity [63] for the repeated measurement analyses.

*p*-values < 0.05 were considered as statistically significant. The used statistical method for the single analyses is indicated in each figure legend.

## 5. Conclusions

In conclusion, our data did not support beneficial effects on obesity and insulin sensitivity described for other *Adcy5*-deficient mouse models. Since effects resulting from Adcy5 reduction were both sex and genetic background-dependent, this may therefore limit the relevance of ADCY5 as potential drug target. Additionally, our *Adcy5^–/–^* model was characterized by a significantly lower voluntary exercise activity in both sexes and diets, suggesting that lower activity may underlie the unexpected phenotype without protection against obesity and its associated cardio-metabolic abnormalities. We find that *Adcy5* plays a role in the regulation of adipocyte size, and smaller adipocytes in *Adcy5^–/–^* mice may be caused by or at least reflect distinct AT gene expression signatures.

## Figures and Tables

**Figure 1 ijms-22-04353-f001:**
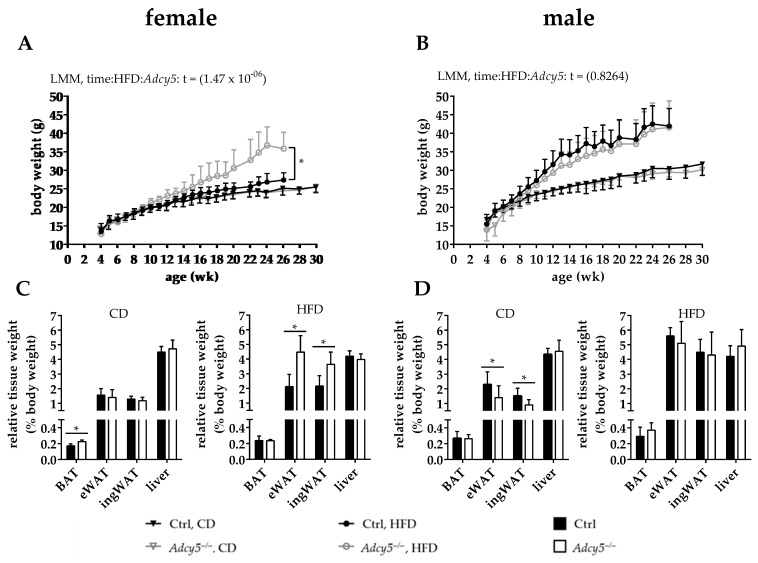
*Adcy5* deficiency did not protect against obesity. (**A**) In females, body weight gain under chow diet (CD) was not different between *Adcy5*^–/–^ and wild-type control (Ctrl) mice, whereas 20 weeks of a high-fat diet (HFD) challenge revealed an inefficient response of Ctrl compared to *Adcy5*^–/–^ mice resulting in a weight gain difference between both genotypes. (**B**) In male mice, both CD and HFD conditions did not reveal differences in body weight dynamics between *Adcy5*^–/–^ and Ctrl mice. Statistical analysis to evaluate the effect of diet and genotype on body weight over time was performed using a linear mixed effect model (Appendix A). Tissue weights of brown adipose tissue (BAT), epigonadal white adipose tissue (eWAT), inguinal WAT (ingWAT) and liver were calculated relative to whole-body weight in female (**C**) and male (**D**) *Adcy5*^–/–^ and Ctrl mice. Genotype-related differences were proofed for statistical significance using multiple t-tests with FDR correction of 5% separately for each diet. Results are expressed as mean ± SD. * Significantly different between Ctrl and *Adcy5*^–/–^ of the same diet with *p* < 0.05. Number of included animals: n (**A**,**C**) = 12 (CD per group), 14 (Ctrl, HFD) and 11 (*Adcy5*^–/–^, HFD). n (**B**,**D**) = 12 (CD, both genotypes), 14 (HFD, Ctrl) and 11 (HFD, *Adcy5*^–/–^).

**Figure 2 ijms-22-04353-f002:**
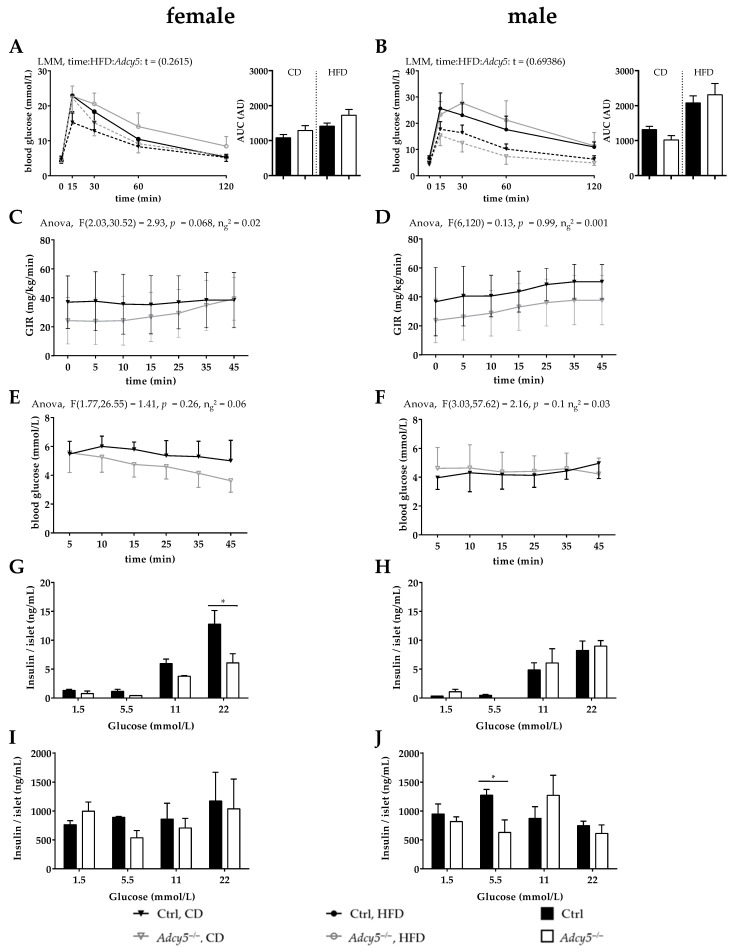
*Adcy5* deficiency (*Adcy5*^–/–^) did not protect against glucose intolerance and insulin resistance. (**A**, left) Compared with control (Ctrl) mice, *Adcy5*^–/–^ females had normal glucose tolerance under chow diet (CD), but tended to develop impaired glucose tolerance after high-fat diet (HFD). Areas under curve (AUC) of intraperitoneal glucose tolerance tests (ipGTT) were not significantly different between female *Adcy5*^–/–^ and Ctrl under both CD and HFD conditions. (**B**, left) Male *Adcy5*^–/–^ mice exhibited lower glucose concentrations 30, 60 and 120 min after the i.p. glucose challenge upon CD, but after HFD this effect was vice versa. Statistical significance to evaluate the effect of diet and genotype on blood glucose over time (**A**,**B**) was calculated using a linear mixed effect model (Appendix A). However, these differences did not translate into significant differences in AUC, after both CD and HFD. In euglycemic-hyperinsulinemic clamps *Adcy5*^–/–^ mice of both sexes (females: **C** and **E**; males: **D** and **F**) under CD conditions were not significantly different in insulin sensitivity as measured by the curves of glucose infusion rate (GIR, **C** and **D**) during the steady state (**E**,**F**). Statistical significance was determined using a two-way ANOVA with *p* < 0.05. Glucose-stimulated insulin secretion (GSIS) of isolated islets (**G–J**). Insulin concentrations secreted into the medium (**G**,**H**) were lower in islets of female *Adcy5*^–/–^ mice upon 2 h stimulation with 11 and 22 mmol glucose (**G**). In males, GSIS was indistinguishable between *Adcy5*^–/–^ and Ctrl mice at all given glucose concentrations (**H**). After glucose stimulation, intrinsic insulin levels of homogenized islets were not different between *Adcy5*^–/–^ and Ctrl mice in both females and males (**I**,**J**). Results are expressed as mean ± SD Statistical significance (*) in **G**–**J** were determined using multiple t-test with *p* < 0.05 and FDR of 5%. Number of included animals: n (**A**,**B**) = 12 (CD per group), 14 (Ctrl, HFD) and 11 (*Adcy5*^–/–^, HFD). n (**C**,**D**) = 12 (CD per group), 14 (Ctrl, HFD) and 13 (*Adcy5*^–/–^, HFD). n (**E**,**F**,**I**) = 8 Ctrl and 9 *Adcy5*^–/–^, n (**G**,**H**,**J**) = 9 Ctrl and 13 *Adcy5*^–/–^.

**Figure 3 ijms-22-04353-f003:**
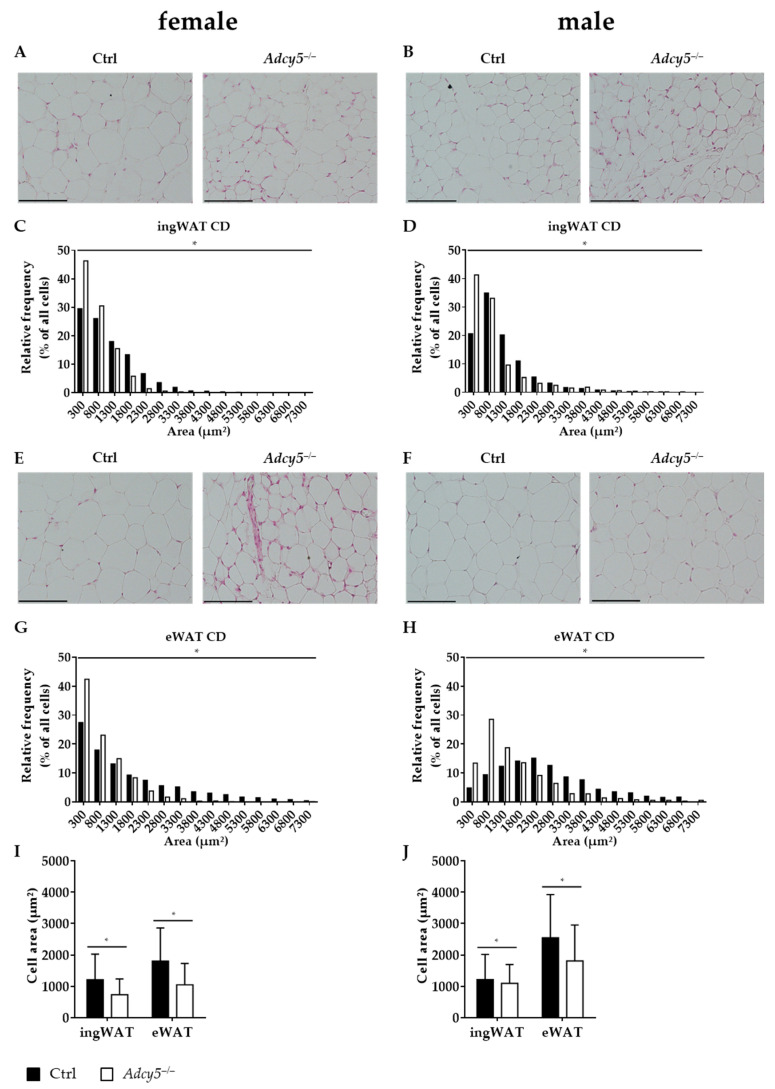
Whole-body *Adcy5* deletion caused smaller adipocytes in white adipose tissue (WAT) after chow diet (CD). Representative inguinal WAT (ingWAT) histology sections of female (**A**) and male (**B**) *Adcy5*^–/–^ and control (Ctrl) mice under CD displayed that *Adcy5*^–/–^ mice have smaller adipocytes. Subfractions of small adipocytes were enriched in female (**C**) and male (**D**) *Adcy5*^–/–^ compared to Ctrl mice. (**E**,**F**) Histological slides from epigonadal WAT (eWAT) confirmed smaller adipocytes in female (**E**) and male (**F**) *Adcy5*^–/–^ mice. Subfractions of small adipocytes were enriched in female (**G**) and male (**H**) *Adcy5*^–/–^ compared to Ctrl mice. Cell area measurements of female (**I**) and male (**J**) *Adcy5*^–/–^ and Ctrl mice revealed lower mean adipocyte size in both ingWAT and eWAT under CD. Scale bar = 100 µm. Data represent mean ± SD. Statistical significance (*) for relative frequencies (**C**,**D**,**G**,**H**) were calculated by using a Chi^2^-test. Statistical significance (*) in **I** and **J** were determined using a multiple t-test with *p* < 0.05 and FDR = 5%.

**Figure 4 ijms-22-04353-f004:**
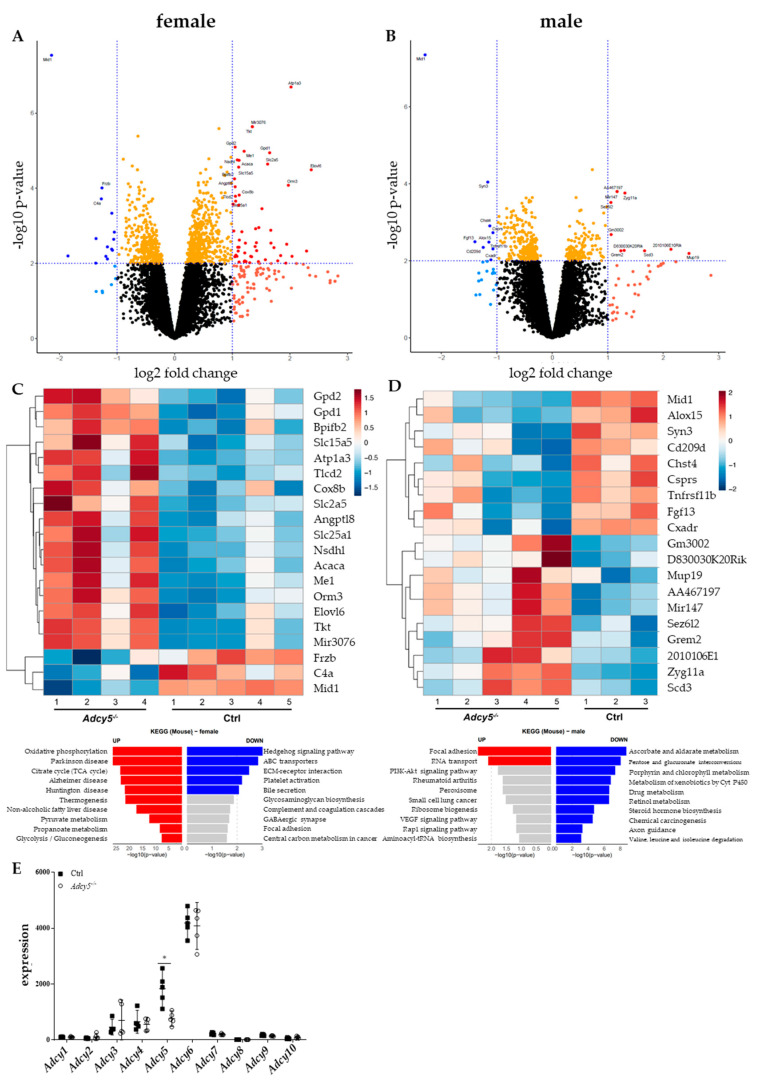
*Adcy5^–/–^* mice had a distinct gene expression signature in epigonadal white adipose tissue (eWAT). Comparison of eWAT gene expression profiles between female (**A**) and male (**B**) *Adcy5*^–/–^ and control (Ctrl) mice at age 30 weeks after chow diet. Microarray gene expression data in the volcano plot are displayed as log2 fold change (FC) vs. the –log10 of the *p*-value. Lower expressed genes in *Adcy5*^–/–^ (log FC ≤ –1; *p*-value < 0.01) compared to Ctrl mice are shown in blue, whereas red color codes higher expressed genes in *Adcy5*^–/–^ (logFC ≥ 1; *p*-value < 0.01). Thresholds are shown as dashed lines. The top 20 genes (sorted by log-odds) are labeled with gene symbols. Top 20 differentially expressed genes in eWAT of female (**C**) and male (**D**) mice are presented as a heat map. Red indicates genes higher expressed in *Adcy5*^–/–^ compared to Ctrl, whereas blue represents lower expressed genes. Additionally, top 10 functional KEGG mouse 2019 pathways of the up- and downregulated differentially expressed genes (DEGs) are provided below the heat map. The vertical axis represents the KEGG pathway terms significantly enriched by the DEGs; the horizontal axis indicates –log10 (*p*-value). Pathways above a threshold –log10 (*p*-value) ≥ 2 are highlighted in red (for upregulated) and blue (for downregulated). (**E**) mRNA expression of adenylyl cyclase types 1–10 in female mice. Only *Adcy5* was significantly lower expressed in female *Adcy5*^–/–^ compared to Ctrl. Data in (**E**) are expressed as single values with mean ± SD. Statistical significance (*) was determined using multiple t-tests with *p* < 0.05 and FDR = 5%.

**Figure 5 ijms-22-04353-f005:**
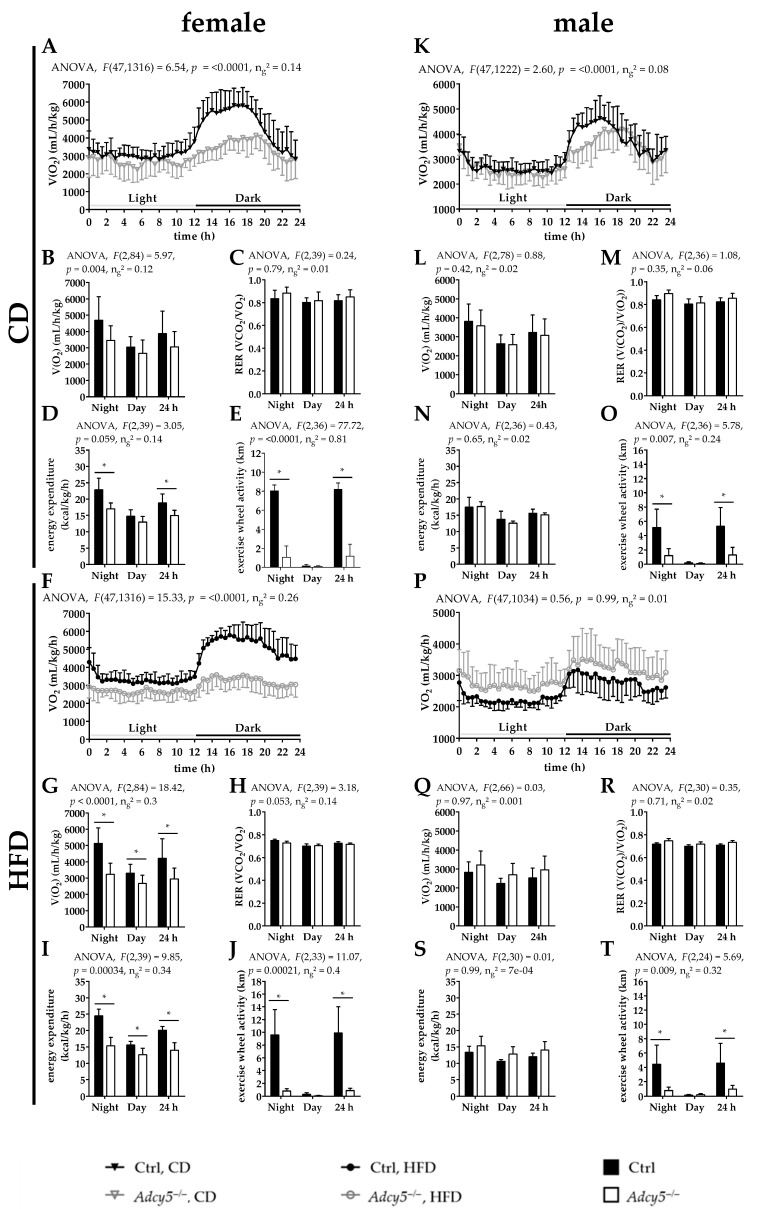
Characterization of energy metabolism in *Adcy5* knockout (*Adcy5*^–/–^) and control mice (Ctrl) under chow (CD) and high-fat diets (HFD). All mice were housed in a metabolic chamber for 72 h. Data are expressed as mean ± SD of two consecutive dark and light phases (in total, 48 h) after a 24-h adaptation phase. Under CD, oxygen consumption (V(O_2_), **A** and **B**) was lower in female *Adcy5*^–/–^ compared to Ctrl mice during the dark phase. Respiratory exchange ratio (RER) was not significantly different between the genotypes in female (**C**) mice under CD. Energy expenditure (**D**) was not, but running wheel activity (**E**) was significantly lower in CD-fed female *Adcy5*^–/–^ mice compared to Ctrl. Under HFD, V(O_2_) (**F**,**G**) was significantly lower in female *Adcy5*^–/–^ mice. RER was not changed in female *Adcy5*^–/–^ under HFD (**H**). Under HFD, energy expenditure (**I**) and running wheel activity (**J**) were lower in female *Adcy5*^–/–^ than in Ctrl mice. (**K**) Male *Adcy5*^–/–^ mice showed partly lower (night) V(O_2_) without significance during mean observation periods (**L**) under CD. RER (**M**) and energy expenditure (**N**) were not significantly different between *Adcy5*^–/–^ and control mice, whereas running wheel activity (**O**) was significantly lower in male *Adcy5*^–/–^ mice under CD. Under HFD, V(O_2_) (**P**,**Q**) was not significantly altered in male *Adcy5*^–/–^ mice. RER (**R**) and energy expenditure (**S**) were unaltered by the loss of *Adcy5* in males. In comparison to females, running wheel activity (**T**) was reduced in male *Adcy5*^–/–^ mice. Oxygen consumption over time (**A**,**F**,**K**,**P**) was analyzed for statistically significant differences between genotypes using a repeated-measured two-way ANOVA. In all bar graphs, statistically significant (*) differences between the genotypes were determined using a two-way ANOVA and pairwise t-tests were calculated for only significant interactions. *P*-values were corrected using the Bonferroni method. Significance was determined by *p* < 0.05. Number of included animals: *n* (**A**–**E**): Ctrl = 6, *Adcy5*^–/–^ = 9, *n* (**F**–**J**): Ctrl = 8, *Adcy5*^–/–^ = 7. *n* (**K**–**O**) Ctrl = 8, *Adcy5*^–/–^ = 6, *n* (**P**–**T**): Ctrl = 5, *Adcy5*^–/–^ = 7.

**Table 1 ijms-22-04353-t001:** Serum concentrations of parameters of lipid metabolism and glucose homeostasis. Measured in female mice at the age of 30 (CD) or 26 weeks (HFD).

	CD			HFD		
	Ctrl	*Adcy5* ^–/–^	n	Ctrl	*Adcy5* ^–/–^	n
**Serum Lipids**						
Triglycerides (mmol/L)	1.07 ± 0.32	1.10 ± 0.20	4 vs. 5	1.03 ± 0.47	1.31 ± 0.22	5 vs. 5
Cholesterol (mmol/L)	**2.16 ± 0.22**	**1.53 ± 0.23**	5 vs. 5	**2.98 ± 0.28**	**3.50 ± 0.30**	5 vs. 5
HDL-cholesterol (mmol/L)	**1.67 ± 0.20**	**1.21 ± 0.20**	5 vs. 5	**2.23 ± 0.27**	**2.78 ± 0.26**	5 vs. 5
LDL-cholesterol (mmol/L)	0.35 ± 0.07	0.18 ± 0.03	5 vs. 5	0.73 ± 0.07	0.58 ± 0.07	5 vs. 5
FFA (mmol/L)	1.50 ± 0.30	1.67 ± 0.35	5 vs. 5	1.71 ± 0.23	1.82 ± 0.26	5 vs. 5
**OLINK Protein serum Analysis (Normalized Protein Expression (AU))**						
Il17f	3.43 ± 0.20	1.84 ± 0.26	3 vs. 4	5.05 ± 0.80	2.79 ± 0.23	3 vs. 3
**Glucose Homeostasis**						
C-peptide (ng/mL)	0.60 ± 0.10	0.40 ± 0.19	5 vs. 5	1.18 ± 0.53	1.11 ± 0.35	5 vs. 5
Insulin (µg/L)	0.28 ± 0.12	0.19 ± 0.03	5 vs. 5	0.69 ± 0.63	0.65 ± 0.48	5 vs. 5
Adiponectin (µg/mL)	208.5 ± 56.1	242.5 ± 105.7	5 vs. 5	140.1 ± 8.3	145.6 ± 4.3	5 vs. 5
Leptin (ng/mL)	3.51 ± 1.12	3.96 ± 5.55	5 vs. 5	**7.02 ± 2.83**	**22.80 ± 10.61**	5 vs. 5
Leptin/body weight (ng/mL/g)	0.15 ± 0.04	0.17 ± 0.24	5 vs. 5	0.27 ± 0.10	0.69 ± 0.31	5 vs. 5
Fasting glucose (mmol/L)	4.57 ± 1.14	4.23 ± 1.34	12 vs. 12	4.96 ± 1.43	6.10 ± 1.85	14 vs. 11
HbA1c (%)	**4.20 ± 0.12**	**3.94 ± 0.18**	12 vs. 11	4.09 ± 0.14	4.09 ± 0.14	13 vs. 11

All values were obtained after a 16-h overnight fasting period. Significantly different values in animals of the same diet are highlighted in **bold**. Statistical significance was determined using a two-way ANOVA with *p* < 0.05. CD—chow diet, HFD—high-fat diet, Ctrl—C57Bl/6NTac, *Adcy5*^–/–^—adenylyl cyclase 5 knockout, HDL—high density lipoprotein, LDL—low density lipoprotein, FFA—free fatty acids, HbA1c—glycated hemoglobin, AU—arbitrary units.

**Table 2 ijms-22-04353-t002:** Serum concentrations of parameters of lipid metabolism and glucose homeostasis. Measured in male mice at an age of 30 (CD) or 26 weeks (HFD).

	CD			HFD		
	Ctrl	*Adcy5* ^–/–^	n	Ctrl	*Adcy5* ^–/–^	n
**Serum Lipids**						
Triglycerides (mmol/L)	1.08 ± 0.09	1.11 ± 0.22	5 vs. 5		1.54 ± 0.10	5 vs. 5
Cholesterol (mmol/L)	2.62 ± 0.49	2.07 ± 0.10	5 vs. 5	**4.30 ± 0.15**	**5.05 ± 0.40**	5 vs. 5
HDL-cholesterol (mmol/L)	2.07 ± 0.33	1.77 ± 0.09	5 vs. 5	**3.28 ± 0.14**	**3.77 ± 0.20**	5 vs. 5
LDL-cholesterol (mmol/L)	**0.39 ± 0.10**	**0.15 ± 0.03**	5 vs. 5	0.83 ± 0.20	1.04 ± 0.15	5 vs. 5
FFA (mmol/L)	1.53 ± 0.09	1.47 ± 0.26	5 vs. 5	1.86 ± 0.08	1.99 ± 0.19	5 vs. 5
**OLINK Protein Serum Analysis (Normalized Protein Expression (AU))**						
Il17f	3.17 ± 0.28	1.56 ± 0.23	3 vs. 4	5.21 ± 1.39	3.14 ± 0.89	3 vs. 4
**Glucose Homeostasis**						
C-peptide (ng/mL)	**1.08 ± 0.41**	**0.45 ± 0.11**	5 vs. 5	2.28 ± 1.37	4.56 ± 1.47	5 vs. 5
Insulin (µg/L)	0.34 ± 0.15	0.17 ± 0.06	5 vs. 5	1.47 ± 1.05	1.81 ± 0.75	4 vs. 5
Adiponectin (µg/mL)	135.2 ± 13.1	107.9 ± 20.7	5 vs. 5	88.3 ± 7.9	85.4 ± 10.7	5 vs. 5
Leptin (ng/mL)	7.69 ± 4.40	2.80 ± 1.26	5 vs. 4	39.8 ± 1.30	40.4 ± 1.17	5 vs. 5
Leptin/body weight (ng/mL/g)	0.25 ± 0.12	0.10 ± 0.05	5 vs. 4	1.00 ± 0.05	0.92 ± 0.02	5 vs. 5
Fasting glucose (mmol/L)	**4.94 ± 1.44**	**3.44 ± 0.55**	12 vs. 12	5.28 ± 2.20	6.18 ± 1.37	14 vs. 13
HbA1c (%)	**4.48 ± 0.15**	**4.29 ± 0.13**	12 vs. 12	4.31 ± 0.17	4.34 ± 0.15	8 vs. 13
HOMA-β (%)	**86.6 ± 38.3**	**190.2 ± 15.3**	5 vs. 5	318 ± 265	281 ± 118	4 vs. 5

All values were obtained after a 16-h overnight fasting period. Significantly different values in animals of the same diet are highlighted in **bold**. Statistical significance was determined using a two-way ANOVA with *p* < 0.05. CD—chow diet, HFD—high-fat diet, CtrlC57Bl/6NTac, *Adcy5*^–/–^—adenylyl cyclase 5 knockout, HDL—high density lipoprotein, LDL—low density lipoprotein, FFA—free fatty acids, HbA1c—glycated hemoglobin, AU—arbitrary units.

## Data Availability

Data is contained within the Appendix A.

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
