# Peer review of "Effects of Whole-Body Adenylyl Cyclase 5 (*Adcy5*) Deficiency on Systemic Insulin Sensitivity and Adipose Tissue"

_ijms, 2021, doi:10.3390/ijms22094353_

Round 1

Reviewer 1 Report

The authors addressed my previous concerns. 

Reviewer 2 Report

I commend the authors for the newly submitted and much more improved version of the manuscript.

This manuscript is a resubmission of an earlier submission. The following is a list of the peer review reports and author responses from that submission.

Round 1

Reviewer 1 Report

This manuscript is weigh much improved to be qualified for the publication in IJMS. 

Author Response

We thank the reviewer for the positive evaluation of our revised manuscript and for recommending it for publication.

Reviewer 2 Report

Please explain the difference between "mice with a whole-body deletion of Adcy5 " and "Adcy5 knockout mice (Adcy5-/-)".

Author Response

We thank the reviewer for raising this point. In our manuscript, we used the terms “whole-body deletion of Adcy5” as a synonym for “knockout” to avoid repeated use of “Adcy5-/-“ or “knockout”. In all cases, whole-body Adcy5 knockouts were used for our experiments. We clarified that in the revised manuscript.

Reviewer 3 Report

This manuscript by Dommel describes the effects of whole body Adcy5 deletion in relation to whole body metabolism, insulin sensitivity and adipose tissue biology. While male Adcy5 KO mice appear indifferent from control mice, female Adcy5 KO mice develop obesity and derived metabolic features compared to control mice when fed a HFD. Moreover, there is a remarkable lack of voluntary activity in the KO mice irrespective of gender. While this phenotype is one of the most pronounced difference between KO mice and their controls, the mechanism for this is not really explored. The strength of the present study is that both male and female mice are studied. However, there are severe methodological concerns that make the results questionable; especially in terms of the statistical handling and the resulting interpretation of the data.

Statistical analyses are currently done mostly be using t-tests, and the risk of making Type 1 statistical errors is certainly present. When you design your study to include two factors such as genotype and diet with two levels in each factor (i.e., KO/Control and CD/HFD), then it is standard practice to conduct two-way ANOVAs. If the same animal is assessed over time, then this factor “Time” should be considered “repeated” and a mixed model ANOVA should be applied. The authors neglect to apply this common way of handling the data, and it is essentially impossible to discern whether the claims made by the authors is true. In all cases where data from the two genotypes on the two diets are presented together in the same figure, the data should be reanalyzed using the appropriate (and completely standard) statistical approaches. Once the authors have done this, the reviewers will be much better able to assess this work.

I have several minor concerns:

  • Please write the Results section in past-tense.
  • Line 103. Please indicate what # indicates in Figure 1E.
  • Fig 1I-L. Please show the blood glucose levels during the clamp. This is necessary to show that the animals are really in steady state.
  • Figure 2C-D+G-H should be analyzed by a Chi-square test (also in Figure S3).
  • Please confirm (and clearly state this in the methods section) whether the p-values for the individual genes in the transcriptomics analysis are FDR corrected.
  • Line 309. Please indicate whether the mice are homozygous for the WT or mutant Nnt allele.
  • The authors appear to use a chow diet as control diet for the HFD (D12492). Please indicate how, or which parameters, these diets are matched.
  • Please add to the methods section that mice KO mice were interbred, so that no littermate control mice could be produced.
  • Please indicate in the methods section the doses of glucose and insulin used in the OGTT and ITT experiments. ITT data appear questionable as insulin does not elicit an expected drop in blood glucose.
  • It would be helpful for the reader if the authors could provide a brief implication summary of their data after each paragraph in the results section. Help the reader get an overview of the results.

Round 2

Reviewer 3 Report

The authors have improved the manuscript, but there are still issues. Especially with the statistical handling of the data. This becomes apparent when the authors claim to have differences between genotypes within a specific diet. To be able to detect difference between genotypes within the treatment, the two factors (genotype and diet) need to interact. If there is no interactions, then it is not ‘allowed’ to perform t-tests (even if these are corrected for multiple testing). Although I do not have access to the raw data (which would have helped in the evaluation), I believe the authors violate this basis statistical principle in the data sets presented in Figure 1D, S2G and S2I. In addition, it does not appear that the authors have used ANOVA testing for the data in Table 1 and 2?! At least the legends says that comparisons were performed by t-tests. Please state the p-value for the interaction effect in those data sets. I need to have clarification on this issue to be able to evaluate the results properly. If no interactions are present (and no main effects of treatment or genotype) then the authors need to update the results section to mirror this.

Another thing is the ITT data. As the dose is now stated in the text, it is quite clear to me that the ITT experiment must have suffered from technical or methodological errors. With the dose of insulin used, there should be a nice drop in blood glucose. This has been reported so many times in the literature. Therefore, I strongly suggest to remove the ITT data (and the discussion of them) and focus more on the clamp data, which is also a better measure of insulin sensitivity.

I think the discussion is overly long and can/should be trimmed by focusing mostly on the major findings from the study. Please also end the discussion with a conclusion/summary paragraph.

Round 3

Reviewer 3 Report

I thank the authors for their response and for providing the raw data from all the experiments. I do not agree that comparisons between diets are not of interest. This is for instance evident from the data provided in Table 1 and 2. The highlighted (in bold) data show difference between genotypes within each diet as determined by a 2-way ANOVA. This is all good. However, in data such as cholesterol and HDL-cholesterol in the female mice, the response in the KO mice compared with the WT mice is opposite in the two diets (decreases in the KOs on the CD, but increases on the HFD). Such interactions between diet and genotype are highly relevant to discuss in the manuscript as they may shed light on the role of ADCY5. The authors fail to recognize this and even say (lines 156-7) that female mice had lower cholesterol with both diets, which is clearly wrong when considering the raw data and the data presented in Table 2.

There is also a need to describe the clamp procedure in more detail since data on insulin sensitivity is central to the paper (those words appear in the title). The referenced paper (Ref 49) does not contain a thorough description of the clamp. For instance, I note from the raw data that GIR at T0 is extremely variable between mice in both WT and KOs. The authors need to better explain how initial glucose infusion rate was determine for each mouse as this is not clear to me. This difference in initial GIR may even affect the results of the clamp.